# Social Distancing and Isolation Strategies to Prevent and Control the Transmission of COVID-19 and Other Infectious Diseases in Care Homes for Older People: An International Review

**DOI:** 10.3390/ijerph19063450

**Published:** 2022-03-15

**Authors:** Sarah Sims, Ruth Harris, Shereen Hussein, Anne Marie Rafferty, Amit Desai, Sinead Palmer, Sally Brearley, Richard Adams, Lindsay Rees, Joanne M. Fitzpatrick

**Affiliations:** 1The Florence Nightingale Faculty of Nursing, Midwifery and Palliative Care, King’s College London, London SE1 8WA, UK; sarah.sims@kcl.ac.uk (S.S.); ruth.harris@kcl.ac.uk (R.H.); anne_marie.rafferty@kcl.ac.uk (A.M.R.); amit.desai@kcl.ac.uk (A.D.); 2London School of Hygiene and Tropical Medicine, London WC1E 7HT, UK; shereen.hussein@lshtm.ac.uk; 3Personal Social Services Research Unit, University of Kent, Canterbury CT2 7NZ, UK; s.e.r.palmer@kent.ac.uk; 4School of Nursing, Kingston University and St George’s University London, London SW17 0RE, UK; sally.brearley@icloud.com; 5Sears Healthcare Ltd., Newbury RG14 1JN, UK; richard.adams@searshealthcare.co.uk; 6Encore Care Homes Management Ltd., Bournemouth BH8 9RL, UK; lr@encorecarehomes.co.uk

**Keywords:** care homes, COVID-19, infection prevention and control, isolation, older people, social distancing

## Abstract

Older people living in care homes are at high risk of poor health outcomes and mortality if they contract COVID-19 or other infectious diseases. Measures used to protect residents include social distancing and isolation, although implementation is challenging. This review aimed to assess the social distancing and isolation strategies used by care homes to prevent and control the transmission of COVID-19 and other infectious diseases. Seven electronic databases were searched: Medline, CINAHL, Embase, PsycINFO, HMIC, Social Care Online, and Web of Science Core Collection. Grey literature was searched using MedRxiv, PDQ-Evidence, NICE Evidence Search, LTCCovid19.org and TRIP. Extracted data were synthesised using narrative synthesis and tabulation. 103 papers were included (10 empirical studies, seven literature reviews, and 86 policy documents). Strategies used to prevent and control the transmission of COVID-19 and other infectious diseases included social distancing and isolation of residents and staff, zoning and cohorting of residents, restriction of resident movement/activities, restriction of visitors and restriction of staff working patterns. This review demonstrates a lack of empirical evidence and the limited nature of policy documentation around social distancing and isolation measures in care homes. Evaluative research on these interventions is needed urgently, focusing on the well-being of all residents, particularly those with hearing, vision or cognitive impairments.

## 1. Introduction

The care home (CH) sector provides care for diverse population groups; the focus of our work is older people, many of whom live with complex and often multiple health needs [1,2]. The CH sector is heterogeneous, but we have used the term ‘care home’ within this work to refer to all long-term care facilities, nursing homes, residential care homes and skilled nursing facilities for older people, which differ substantially in their case mix, skill mix and staffing ratios. SARS-CoV-2, also known as COVID-19, is a rapidly emerging infectious disease [3], and healthcare setting transmission plays a vital role in its spread [4,5]. Older people living in CHs are at high risk of poor health outcomes and mortality if they contract COVID-19 [6]. Therefore, CHs have implemented measures such as social distancing and isolation to protect them from the disease. The CH sector has stated that implementing social distancing and isolation when caring for residents is a significant challenge [7]. The evidence base to support the delivery of social distancing and isolation in CHs is lacking [6]. This rapid review aims to identify social distancing and isolation measures previously and currently used by CHs to prevent and control the transmission of COVID-19. Social distancing and isolation measures used to prevent and control the spread of other infectious diseases in CHs are also included so that there is an opportunity to learn from the evidence in these areas (PROSPERO registration: CRD42021226734). Our review has been undertaken as part of a National Institute for Health Research, Health and Social Care Delivery Research (NIHR HS&DR Project no: 132541) funded study that aims to explore and understand the real-life experiences of social distancing and isolation measures in CHs for older people from the perspective of multiple stakeholders [8]. 

## 2. Materials and Methods

### 2.1. Review Design and Conceptual Basis 

We conducted a review of published literature on social distancing and isolation as measures to prevent and control the transmission of COVID-19 and other infectious and contagious diseases in CHs for older people. A rapid review methodology was selected due to the time-critical nature of the ongoing pandemic, and we followed the Cochrane rapid review guidance [9]. 

### 2.2. Review Questions, Boundaries and Scope

This review aimed to identify and assess the social distancing and isolation strategies previously and currently used by CHs to prevent and control the transmission of COVID-19 and other infectious diseases. Specific questions were: What mechanisms and measures have been used to implement social distancing and isolation for residents and staff?How are they implemented? What are the challenges and facilitators to implementation?What is the impact of the implemented measures and mechanisms?
What are the psychosocial and physical consequences for older people? What are the consequences for family members, staff, and organisations? What is the evidence of measures and mechanisms that work for different CHs and resident needs? What recommendations have been made after the implementation of these measures? 



### 2.3. Literature Searching and Analysis 

The search strategy was developed in consultation with Information Services Specialists at (name of institution blinded for review):“nursing home* OR care home* OR long-term care* OR long term care* OR aged care facilit* OR aged-care facilit* OR residential care home* AND infect* control* OR infect* prevent* OR cohort* OR zon* OR quarantin* OR social distanc* OR prevent* OR isolat* AND acute respiratory infection* OR clostridium difficile* OR diarrhoea OR vomit* OR methicillin-resistant staphylococcus aureus* or SARS* OR MERS-CoV* OR flu* OR SARS-CoV19 OR SARS-CoV-2 OR COV* OR Corona* 

This search strategy was run on 13 January 2021 in seven electronic databases: (Medline, CINAHL, Embase, PsycINFO, HMIC, Social Care Online, and Web of Science Core Collection) and a total of 4753 papers were identified. Grey literature relating to policy and organisational-based material was also searched between 20–24 January 2021 (MedRxiv, PDQ-Evidence, NICE Evidence Search, LTCCovid19.org and TRIP) and 13,488 articles were identified. After removing the 1465 duplicates from these 18,241 documents, 16,776 articles remained, and the titles and abstracts were screened independently by two reviewers using the inclusion and exclusion criteria. These were: the literature needed to address COVID-19 or other infectious and contagious diseases in older people (aged 65 years and over) living in CHs, nursing homes, long-term facilities, or residential CHs. No limits were placed on the geographical location, but only English-language articles were included because of the resources available. 145 abstracts were identified as potentially relevant, and these papers were independently reviewed in full by four reviewers using the inclusion and exclusion criteria to make a recommendation: ‘Include’; ‘Exclude’; ‘Unsure—need to conduct full text screening’ (each paper was reviewed by two reviewers and any conflict in the assessments were resolved in collaboration with a third reviewer). 94 records were included in the review. Targeted searching of the reference lists of these 94 papers highlighted a further 10 papers, which were again reviewed independently by two reviewers, and nine were included in the review. Thus, a total of 103 papers were included in the review [6,7,10,11,12,13,14,15,16,17,18,19,20,21,22,23,24,25,26,27,28,29,30,31,32,33,34,35,36,37,38,39,40,41,42,43,44,45,46,47,48,49,50,51,52,53,54,55,56,57,58,59,60,61,62,63,64,65,66,67,68,69,70,71,72,73,74,75,76,77,78,79,80,81,82,83,84,85,86,87,88,89,90,91,92,93,94,95,96,97,98,99,100,101,102,103,104,105,106,107,108,109,110] (see Table 1 for a description of each paper), and 52 papers were excluded [111,112,113,114,115,116,117,118,119,120,121,122,123,124,125,126,127,128,129,130,131,132,133,134,135,136,137,138,139,140,141,142,143,144,145,146,147,148,149,150,151,152,153,154,155,156,157,158,159,160,161,162]. 

Data were extracted from the 103 included papers into a bespoke extraction form, using an Excel spreadsheet, which was reviewed and tested within the team. Data items included: author(s) and year of publication; study aim; study design; setting and participants; intervention(s) discussed, including a description of the measure(s) used (e.g., what it was; who it was for; how it was implemented, factors supporting or hindering its implementation); findings; and author recommendations. Findings from the 103 papers were synthesised using tables and a narrative summary. The narrative synthesis was organised around the review questions: interventions for the prevention and control of COVID-19 and their impact; challenges and facilitators for implementing COVID-19-related interventions in care homes; and interventions for the prevention and control of other (non-COVID-19 related) infectious diseases. Figure 1 highlights a flowchart of the review process.

## 3. Findings

The 103 papers consisted of 10 research studies, eight of which explored COVID-19 and two explored other infectious diseases. Two empirical studies mentioned social distancing interventions [90,99], nine mentioned isolation interventions [17,32,54,58,81,86,90,97,99], eight mentioned restrictions [17,18,58,81,86,90,97,99] and two mentioned zoning or cohorting [90,99]. However, it should be noted that these interventions were generally mentioned as part of a wider discussion of COVID-19 strategies and were not the main focus of the studies. Three of these empirical studies were conducted in the UK, four were conducted in Europe, two in Asia and one in North America. The design of the 10 research studies was heterogenous. Seven different quality assessment tools were used by a single reviewer to conduct a risk of bias assessment, with verification of judgements by a second reviewer: from the Critical Appraisal Skills Programme (CASP) checklists for RCTS, qualitative research and cohort studies [164]; the mixed methods appraisal tool (MMAT) [165]; from the Joanna Briggs Institute (JBI) checklists for analytical cross-sectional studies and prevalence studies [166]; and Jungers (2017) guidance for reporting on Delphi studies [167]. There was reviewer agreement that all 10 studies should be included in the review.

Also included in this review were 86 policy documents/grey literature and seven literature/rapid reviews. Grey literature came from around the world and included policy documents highlighting different countries’ responses to the pandemic, guidance/guidelines for CHs, briefing documents, discussions, and commentaries. The included literature/rapid reviews were of varying quality (some were pre-print and not peer-reviewed), with five related to COVID-19 and two related to other infectious diseases. All papers highlighted the various strategies used by CHs to prevent or control the transmission of COVID-19 and other infectious diseases amongst their residents and staff, including not only strategies of social distancing and isolation, but also restrictions, zoning and cohorting. However, once again, the focus of grey literature and reviews was generally on overall responses to the pandemic, with these specific interventions mentioned as one component of this. Other strategies, such as the use of personal protective equipment (PPE), testing, ventilation, and adequate hygiene procedures, were also highlighted but are outside the scope of this review. 

### 3.1. Interventions for the Prevention and Control of COVID-19

The following interventions were discussed specifically in relation to the spread of COVID-19:

#### 3.1.1. Social Distancing

The terms ‘social distancing’ and ‘physical distancing’ were used interchangeably within and across papers, but for purposes of consistency they are referred to as ‘social distancing’. There was little discussion of social distancing interventions within CHs. The term was not defined in the literature or what it meant in practice, other than that, CHs must adhere to ‘government guidance’ or ‘national rules’ on distancing. Those who did describe their understanding of distancing stated this was maintaining a distance from other people of at least one to two metres in Europe or six feet in the US [19,24,46,57,75,104,107]. 

#### 3.1.2. Social Distancing for Residents 

Social distancing generally referred to those residents who had not been exposed to COVID-19 being able to continue with some regular routines and group activities whilst maintaining a distance from others [24,34,37,40,47,51,57,62,68,75,82,83,90,99,104,107,110]. This included socially distanced mealtimes in the dining room [47,51,62,99,107]; separated chairs in common rooms [51]; one-way movement systems around the home [41]; and spacing indicators on the floors [51]. Some papers stated that social distancing measures enabled residents to maintain a “normal” life during the pandemic [24]. However, other homes decided not to enforce social distancing measures, knowing that their residents would not be able to adhere to them [62]. Very little was stated about the impact of social distancing measures on residents other than acknowledging that they may have severe implications for their mental health and wellbeing [107]. Residents with cognitive impairment or dementia were also reported to have greater difficulty understanding and abiding by social distancing measures [6,94]. 

#### 3.1.3. Social Distancing for Staff 

Some papers discussed the importance of social distancing among staff, for example, in staff rooms and other areas around the CH [16,46,51,73,75,107,110], including separating chairs in staff rooms [51], staggering breaks to limit the density of staff in specific areas [57], and restricting staff car sharing to/from work [57]. No reports of the impact of social distancing on staff were identified. 

#### 3.1.4. Isolation

The terms ‘quarantine’ and ‘isolation’ were used interchangeably within and across papers, but for consistency are referred to as ‘isolation’:

#### 3.1.5. Isolation of Residents 

Some CHs cared for all residents as though they were COVID-19 positive, which meant isolating everyone within their private rooms regardless of their COVID-19 status [24,58,81,110]. However, others only asked residents to isolate if they had suspected COVID-19 symptoms, if they had come into contact with someone with COVID-19 or if there was an outbreak within the care home [12,14,16,28,30,33,36,37,39,40,45,47,51,52,57,58,62,63,67,72,75,85,87,93,96,97,100,102,103,104,105,106,107,108,109,110]. Isolation was also implemented when new residents entered the CH [6,19,29,33,39,40,51,56,62,68,87,90,100,107,108,110] or when residents were discharged from hospital [6,10,20,29,33,34,38,39,40,51,56,57,62,68,72,75,87,102,107,109,110]. Globally, isolation was often required for 14 days [16,37,39,52,57,107,110], though some papers from Europe, South America and Asia highlighted requirements of 10 days [36,106,110]. Where possible, residents with COVID-19 were to be isolated in their own bedroom [12,14,16,19,28,33,40,47,51,57,72,75,85,97,103,104,105,106,107,110] or transferred to a hospital, ‘sanitary house’, hotel or other community setting, where available/applicable [10,13,20,33,52,56,103,109]. If possible, isolation rooms should have an en-suite bathroom, but where this was not available, a dedicated bathroom/commode should be identified [40,51,57,75,85,104,105]. 

Two policy papers stated that where isolation was not undertaken effectively, the virus spread amongst CH staff and residents [13,47]. One (pre-peer review) empirical study [86] also reported higher odds of outbreaks in CHs with poor compliance with isolation procedures. However, there was evidence from empirical research, literature reviews and policy documents that the isolation of residents could have negative effects upon their physical and mental health [6,12,24,34,48,62,90,94,97,103,106]. This was particularly notable for those with dementia, cognitive problems, autism, and learning difficulties, who might not fully comprehend instructions [15,34,38,62,68,90,94,107]. For these individuals, agitation and behavioural disturbances were reported [15,48,107], and one commentary paper stated this may have required the increased use of restraint [15]. Isolation has also been associated in the grey literature with decreased movement and mobility [24,47,106,107]; increased postural disorders [24]; increased risk of falls [24]; and increased sarcopenia and deep vein thrombosis [19]. Isolated CH residents have been reported in empirical research and policy documents to have poorer fluid/food intake, leading to weight loss, malnutrition and difficulties maintaining hydration [24,26,48,81,88]. 

#### 3.1.6. Isolation of Staff 

Isolation measures were implemented for CH staff, such as an isolation period for those returning from a hospital stay [87,99] or international travel [72,102]. Isolation was also required for those staff who had COVID-19 symptoms or who had contact with someone with COVID-19 [14,19,33,36,37,38,51,52,57,62,68,75,107,110]. Usually, the isolation period for staff was 14 days [16,33,52,56,68,75], but in some cases staff could return to work after ten [57] or seven days [16,68]. Grey literature stated that isolation guidelines for staff could adversely affect CHs by creating significant staff shortages [21,33,67], and there were reports of some homes experiencing dilemmas around this. For example, there were accounts in the US of staff who had come into contact with COVID-19 being asked to continue working if they did not display symptoms themselves [75] and in the Netherlands, some CH staff were asked to keep working while sick [21,67]. There were also examples from New Zealand of residents being transferred to hospitals due to insufficient staff available to care for them [72].

#### 3.1.7. ‘Zoning’ and ‘Cohorting’

The terms ‘zoning’ and ‘cohorting’ are understood differently and were used interchangeably across papers, with some also discussing ways of separating residents without using any specific terms to describe these interventions [6,7,12,18,19,27,28,29,33,35,36,38,42,46,51,52,60,63,65,68,72,75,77,81,83,84,85,88,90,96,106,107,109,110]. For this review, ‘*zoning*’ refers to creating *physical separation areas* within a CH, for example, separating residents with and without COVID-19 onto separate floors or disparate wings. The term ‘*cohorting*’ refers to all other imposed means of grouping residents, including allocating specific groups of residents to particular areas within a floor. 

Empirical research papers, literature reviews and policy papers referred to ‘zoning’ residents with a positive COVID-19 test result/suspected COVID-19 away from those without [6,12,19,27,29,33,35,36,42,46,52,68,81,84,88,90,109,110]. The intervention of zoning was reported to offer CHs a clear delineation of risk zones throughout the building and it was stated that staff, residents, and equipment should not move between the zones to reduce cross-contamination [12,19,29,33,41,46,52,110]. Separate staff entrances, exits and corridors for each zone were used, where possible [19,41,46,52,90,99], with staff communicating via telephone [41,52]. This intervention allowed CH zones to operate as self-sufficient care bubbles [52], enabling residents to have limited freedoms within their zone [6,27,42,81], encouraging socialisation and activity within the zones and helping decrease residents’ feelings of isolation and loneliness [90]. Cohorting was sometimes suggested in the grey literature for settings where it was impossible to physically separate residents [19]. Examples of cohorting were organising residents into small groups/dedicated areas within a floor (rather than separate floors or wings) with the same staff continuously assigned to them [19,41,51,58,63,75,96,106,107,109]. The rationale for this was that, in case of infection within this small group, as few residents and staff as possible would require isolation [58]. 

There was some empirical research evidence that staff working solely within an allocated zone/cohort of residents helped prevent the spread of COVID-19 [86]. However, such interventions depend upon CHs having sufficient staff resources [90] and can result in financial costs [41]. Indeed, examples in the grey literature were provided where zoning and cohorting interventions required CHs to recruit new team members or to use ‘surge staffing’ (i.e., pre-identified temporary, casual labour) [36,41]. There was some evidence from empirical research and the grey literature that moving residents from their standard room to a new cohort/zone could create confusion, anxiety, or distress for residents [40,90]. Still, it was acknowledged that, for many, the benefits were likely to outweigh the negative consequences [90]. 

#### 3.1.8. Restrictions 

Although this review focussed upon social distancing and isolation interventions, other related interventions of ‘restrictions’ were also regularly discussed. For this review, the term ‘restrictions’ referred to any instances where an individual was *prevented* from doing something they would normally do (e.g., cancelling activities) or asked to *modify* how they would typically do something (e.g., asking residents to eat meals in their bedroom). We have separated this from ‘social distancing’ (which referred to instances where an individual could carry on activities of everyday life whilst remaining at a distance from other individuals) for the purpose of clarity, but these terms were used interchangeably within the literature. 

#### 3.1.9. Restrictions Placed upon Residents 

Several restrictions were reported for CH residents, including staggering mealtimes or serving meals in residents’ rooms rather than communal dining rooms [18,31,33,58,75,83,102,106]; preventing residents from leaving the home [10,12,97] and reducing/cancelling group activities [18,24,30,31,33,36,44,51,83,97,99,106,108]. There was no empirical evidence on whether implementing resident restrictions had any impact upon preventing the transmission of COVID-19 in CHs. Still, there was evidence from empirical research, literature reviews and policy documents that restrictions could impact residents’ health and wellbeing, including physical, cognitive, psychological, and functional declines [15,18,26,30,34,36,48,51,58,62,63,67,69,73,81,91,94,97,103,106,107]. There were reports of reduced fluid and food intake [81], and increased agitation, behavioural disturbances, anxiety, and psychotropic medication use [15,35,62,69,81,91,107]. 

#### 3.1.10. Restrictions Placed upon Staff 

Several papers also talked about the restrictions placed upon staff members [17,86]. These restrictions involved changes to working patterns, such as shift length, rota patterns, and extended working hours [10,24,42,51,57,70]; limiting the number of settings staff could work within [33,40,42,52,61,83,110]; and asking staff to live within the CH for extended periods [17,40,53,77,109,110]. Policy papers stated that professional practices were redefined, with tasks modified and adapted to suit new working rhythms and procedures, such as sorting bedding, disinfecting premises and serving meals [24]. In the US, staff training and certification requirements were modified to reinforce the available workforce [31,55], whilst Australia, New Zealand and Malaysia increased the maximum weekly working hours allowed by international students and those with restricted work visas [33,56,72]. Slovenia restricted CH staff’s right to leave their employment or strike [33]. There was some empirical evidence that staff confinement in CHs could be beneficial for transmission rates, though only one research study explored this [17], alongside anecdotal evidence in a news article [53]. There was, however, suggestion from literature reviews and policy documents that new ways of working and more significant staff absences increased workloads and led to stress, exhaustion and burnout [24,27,34,35,48,62,69,103]. 

#### 3.1.11. Restrictions Placed upon Visitors 

In most countries, families and friends were prevented from entering CHs (particularly in the first wave of the pandemic), other than in end-of-life/special circumstances [6,10,12,13,16,17,18,20,21,24,25,28,29,30,31,33,34,35,36,40,42,44,46,47,52,56,57,60,61,63,68,70,72,77,81,83,84,85,87,88,90,91,93,94,95,96,97,99,100,102,103,107,108,109]. Several innovative interventions were adopted to provide alternative ways for families to maintain contact with residents during these periods of restriction. These included: window/car/garden visits [12,26,30,35,36,42,46,47,51,62,63,69,70,81,90,91,94,96,108,110]; the installation of tents, glass pods or plexiglass containers/walls [46,63,94,97,108,110]; and the setting up of telephone/video calls [28,30,46,56,63,70,75,88,90,96,99,102,103,104,105,106,107,108,109]. When family members *were* able to enter the CH (e.g., at a resident’s end of life or when lockdowns were eased), varying restrictions remained in place, including limits on the number and/or duration of visits [24,25,26,33,38,60,61,68,69,73,90,104,110]; ensuring visits were supervised [69]; and using designated visitor entrances/exits/reception rooms [16,26,45,46,51,90,99,110]. 

There was evidence from the empirical research, literature reviews and policy documents that visitor restrictions negatively impacted residents’ health and wellbeing, with residents reported to be confused, distressed, and frustrated by not seeing their family [10,35,48,58,62,88,91,94,97,100,103,106,108]. In one research study, residents did not recognise their families after restrictions had eased [97]. These restrictions also negatively impacted the well-being of families, with reports of guilt, fear, worry and isolation [26,34,47,60,69,73,90,96,103,107]. Some residents died without having their family with them [58,60,63,91,96], which was distressing for families [34,60,90,96] and staff [60,90]. One policy paper stated that some CHs experienced financial difficulties when older people expressed reluctance to move into them due to fears they would be isolated from their families [21]. 

#### 3.1.12. Restrictions Placed upon Other Professionals and Services 

In many instances, all but essential professionals/services were restricted from entering CHs during the COVID-19 pandemic. This included healthcare professionals (e.g., physicians, psychologists, physiotherapists etc.) and non-healthcare workers, (e.g., hairdressers, entertainers, and volunteers) [17,18,36,40,72,75,96,97,102,110]. Generally, healthcare services moved towards virtual or remote ways of working, including video calls/consultations and virtual ward rounds/multidisciplinary team meetings [40,62,70,90,99,102]. There were some concerns in the grey literature that restricting professionals from entering CHs may have prevented residents from receiving necessary medical and social care [45,95,107]. Concerns have also been raised in the grey literature about the quality of care provided to residents during periods of restrictions, as external regulators did not enter CHs to undertake inspections for quality assessments or advisory visits [30,33,48,55,70,95]. 

Table 2 highlights the challenges and facilitators reported impacting the success of implementing COVID-19 interventions in CHs.

### 3.2. Interventions for the Prevention and Control of Other (Non-COVID-19 Related) Infectious Diseases

Twenty papers discussed strategies used by CHs to prevent and control the transmission of infectious diseases and healthcare-associated infections (HAIs) other than COVID-19 [22,23,32,43,49,50,54,59,64,66,71,74,76,78,79,80,89,92,98,101]. These included influenza, urinary tract infections (UTIs), respiratory infections, pneumonia, methicillin-resistant Staphylococcus aureus (MRSA), gastroenteritis, multidrug-resistant organisms (MDROs) and Clostridium difficile (C-diff). *Surveillance* was the strategy most commonly discussed as being used by CHs to *prevent* infectious diseases from entering the premises [54,74,76,80,89,92,98,101]. Surveillance involved the systematic collection, consolidation, and analysis of data related to infectious diseases, which aimed to ensure the early identification of symptoms [74]. Information on the implementation of social distancing and isolation measures, restrictions, zoning and cohorting for infectious diseases other than COVID-19 was more limited. 

#### 3.2.1. Social Distancing 

Only two papers from the grey literature discussed social distancing measures for non-COVID-19 related infectious diseases [80,98]: one stated there should be at least two metres between residents with and without signs and symptoms of influenza [80] whilst the other said that CHs should maintain a one-metre distance between all residents during outbreaks of respiratory infection [98]. 

#### 3.2.2. Isolation

Several papers discussed how isolation had been used to control the spread of other infectious diseases within CHs, such as MRSA, influenza, and C-diff [22,23,32,43,49,50,59,64,71,76,78,79,80,89,98]. This tended to involve isolating infectious residents within their bedrooms or cohorting them where this was not possible [22,23,49,50,64,71,74,76,78,79,80,89,98]. Some papers talked of the need to restrict admissions of new residents into the CH and/or prevent the readmission of those who had been in hospital during severe outbreaks [23,74]. Others highlighted a more flexible approach to isolation measures than there was for COVID-19. For example, one research study noted that known MRSA carriers were only asked to be separated from vulnerable residents with skin lesions or indwelling catheters, but were otherwise allowed to continue with usual social activities [32]. The importance of making decisions around isolation on a case-by-case basis was emphasised in the grey literature, as was the importance of not over-isolating residents [22,43,79,89]. There were also examples highlighted in the grey literature of staff being asked to isolate to control the spread of influenza by self-monitoring for symptoms of illness and staying away from work if feeling unwell [23]

#### 3.2.3. Restrictions

Only one literature review mentioned the use of restrictions for preventing infectious diseases from entering CHs [66] and stated there was no evidence to support banning/restricting visits to CHs for this purpose. More papers explored how restrictions could control the spread of infectious disease when there was already an outbreak or suspected case within the CH. Some policy documents reported that restricting the movement of residents and visitors during an outbreak of infectious disease could be beneficial [23,50,71,74,80,89,92,98]. This included the restriction of group activities in addition to minimising the movement of visitors within the CH [23,74,80,89,98]. A Canadian toolkit reported that complete closure of CHs to visitors should not be permitted unless under the order of the Medical Officer of Health, due to the potential harm this could cause residents and families [74]. There was also less of a ‘blanket approach’ to restrictions reported in policy documents for other infectious diseases than for COVID-19. These included, for example, residents only being restricted from group activities when wound drainage or diarrhoea could not be contained [71] or activities only being restricted for residents in outbreak ‘zone’ areas [74]. Again, it was reported that clear signage and communication were important for residents and family members during any periods of restriction [98]. Some policy documents also discussed the restrictions that should be placed upon staff working patterns to control the spread of infectious diseases in CHs [23,74,89]. For example, during influenza outbreaks, a policy paper [23] reported that the movement of staff across CHs and healthcare facilities should be minimised. Similarly, where zoning/cohorting restrictions were in place, staff working within affected units should not also work within non-affected areas of the home [23,74,98]. Finally, it was suggested in the grey literature that only staff who had been vaccinated against influenza should care for those residents with suspected/confirmed influenza [23,74].

Similar challenges and facilitators impacting the success of implementing interventions for other infectious diseases in CHs were identified as for COVID-19 interventions, including the need for staff education and training around infection control measures [74,101], the design and layout of CH buildings [50,64] and good communication with residents and visitors [64].

## 4. Discussion and Conclusions

This review demonstrates the overall lack of empirical evidence and the limited nature of documentation around social distancing and isolation measures in CHs. Most papers identified within this review were grey literature or policy documents, many of which were descriptive or opinion based. Furthermore, these interventions were generally mentioned as part of a wider discussion of COVID-19 strategies and were not the main focus of the papers. Even fewer papers discussed these measures for non-COVID-19 related infections, which meant learning from this evidence base was also limited.

This review identified limited exploration of social distancing interventions to prevent and control the spread of COVID-19 in CHs. Up to the date of this review, only 10 empirical studies met the inclusion criteria, and only two discussed the impact of social distancing in CHs. This review addresses this gap and contributes to a body of research evidence that is now developing apace. Of significance is the plethora of policy documents on the topic. The grey literature provided little evidence of the effects of social distancing on resident outcomes or COVID-19 infection rates and no discussion of the impact of social distancing on CH staff. More literature explored isolation interventions in CHs, including nine empirical studies, and a key finding was the considerable variation in available guidelines and the implementation of measures, nationally and internationally. This reflects the challenges for CHs of dealing with rapidly changing multiple sets of guidance [168,169], the CH sector being ill-prepared to cope with a pandemic [168,170] and the sector not being supported well at the outset, with reports of abandonment by governments [168,171,172,173]. These findings contribute to important lessons for decision makers about the need for comprehensible, concise, and meaningful guidance about social distancing and isolation and related measures that can be translated easily into operational policies for care homes. CHs need evidence-informed guidance that sets out what and how social distancing and isolation measures should be operationalised, whilst meeting residents’ individual needs, including their fundamental rights to liberty and security, and with attention to education for residents, families, friends and staff [170]. Further, there is a need for large, evaluative, empirical studies about the impact of social distancing and isolation measures on the populations of older people living in CHs worldwide.

The most discussed intervention for preventing and controlling the transmission of COVID-19 and other infectious diseases in CHs was restrictions, which included restricted visiting from families or friends and external agencies, restricted group events and activities for residents, and restricted work arrangements for staff. There was limited empirical evidence on whether visitor restrictions prevented the transmission of COVID-19 and other infectious diseases in CHs. Thus, many authors have highlighted the importance of reintroducing visitors into CHs as soon as it is safe to do so [40,58,107], with calls for CHs to take a more flexible, risk-assessment-based approach to visits [38,67]. The urgency of this is accentuated by evidence emerging about the negative impacts of visiting restrictions on the physical, psychological, emotional and cognitive well-being of residents and their families and friends [174,175]. These findings resonate with other studies for nursing home residents [168], where restrictions resulted in several losses related to freedom, social contact, activities, communication and autonomy, and with residents describing feelings of depression, hopelessness, uselessness, and sadness. The implementation of any infection control and prevention measures must prioritise the well-being of all residents, with targeted consideration around how best to achieve this for residents living with particular care needs, such as hearing, vision or cognitive impairments.

Many CH residents live with dementia; in UK CHs, the prevalence of dementia is 69% and increasing [176]. Restrictions on the movement of residents living with dementia have been shown to have negative consequences for mental wellbeing, with an escalation of neuropsychiatric and behavioural disturbances [177]. A recent rapid systematic review of the effect of COVID-19 isolation measures on the cognitive and mental health of people living with dementia included only two studies that had been conducted in CHs [178]. Findings reported worsening of memory and mood, and reduced independence in activities of daily living. There was also an increase in mean depression and anxiety scores. Further research is required to understand more fully the experience of infection prevention and control measures, including social distancing and isolation for older people living with cognitive impairment, their families, friends and staff, to inform evidence-based practice that maximises quality of life and well-being.

Several factors were identified as supporting CHs in their implementation of interventions to control the transmission of COVID-19 and other infectious diseases, including access to innovative technology [60,90,99]; good communication with residents and families [40,64,75,104]; and ensuring CH staff were sufficiently trained and supported [42,52,61,74,101]. These findings concur with a recent review [179] to analyse the impact of COVID-19-related social distancing requirements on older adults living in long-term care facilities. Strategies proposed to mitigate the negative effects of social distancing were: the use of technology; maintaining virtual intergenerational connections; maintaining therapeutic and personalised care; and adhering to COVID-19 safety guidelines and preventive measures [179]. A coherent, agreed strategy is pivotal to support the implementation of these action points nationally and internationally. A lack of guidance and clarity from governments around when and how interventions should be applied was also identified as a potential barrier, with policy measures often scarce, flawed, or adopted late [81,95,108]. Again, nursing home staff in a recent study [168] shared that early in the pandemic, information and instructions about what to do and when were unclear, sometimes incoherent and ever-changing. CH staff have responded innovatively to the challenges of implementing social distancing and isolation measures in adverse circumstances, coping with additional workloads and resource constraints [172]. There is a need for key stakeholders, including researchers, funders, the CH sector, and governments to understand fully their experiences of actions that worked, did not work, or worked less well, and why, and to work collaboratively with CHs to ensure that their staff are supported and enabled to care well for residents and their families and friends for the duration of this pandemic and beyond.

A further barrier to social distancing and isolation interventions identified by this review was the design of CH buildings. Many CHs have insufficient space to provide single isolation rooms, create separate ‘zones’ or ensure sufficient walking space around the home in line with social distancing measures [90,103,109]. This is a significant issue that warrants careful discussion and planning. There have been calls for new minimum standards for the design of UK CHs so that they can respond effectively to any future outbreaks of infectious diseases whilst promoting quality of life and well-being for residents, their families, friends, and staff (https://www.buildingbetterhealthcare.com/news/article_page/Call_for_new_minimum_standards_for_UK_care_home_design/167833) (Access on: 24 November 2021). ‘Resilient building design’ for CHs that addresses design for infection control as well as for improved quality of life has been recommended [11]. It is acknowledged that this is complex and multifaceted, and will evolve as new CH facilities are purposefully designed and built.

Notably, this review has contributed to clarifying terminology related to the concepts of social distancing and isolation as infection prevention and control measures within the context of COVID-19 and other infectious diseases. A key finding was inconsistency in the meaning and use of key terms such as ‘social distancing,’ ‘isolation’ and ‘restrictions,’ and such inconsistency across guidance, protocols and policies needs to be addressed.

### 4.1. Strengths and Limitations

This is an important topic and our review makes an important contribution to understanding in the field. To our knowledge, this is the most extensive review of the evidence around social distancing and isolation measures to prevent and control the transmission of COVID-19 and other infectious diseases in CHs caring for older people. Cochrane rapid review methodology was used, contributing to the quality of conduct and reporting and the robustness of the results. Our systematic and comprehensive searches of several databases and the grey literature to answer the review questions culminated in the inclusion of 103 records. The 103 papers were from around the world. Only sources published in English were included, which is acknowledged as a potential source of publication bias. Other key strategies in the prevention and control of COVID-19 and other infectious diseases, such as the use of PPE, testing, ventilation and adequate hygiene procedures, were excluded as they were outside the scope of this review, and we acknowledge this is a limitation.

### 4.2. Conclusions

The COVID-19 pandemic has had a devasting impact on the CH sector, and in many countries, CHs have been at the epicentre of deaths from the disease [18,81,86,99]. To help prevent and manage COVID-19, our review has advanced understanding of social distancing and isolation for older people living in CHs. The empirical phase of our study will contribute to understanding further the real-life experiences, challenges, facilitators and consequences of implementing social distancing and isolation within the CH setting, informing best practice guidance and resources.

## Figures and Tables

**Figure 1 ijerph-19-03450-f001:**
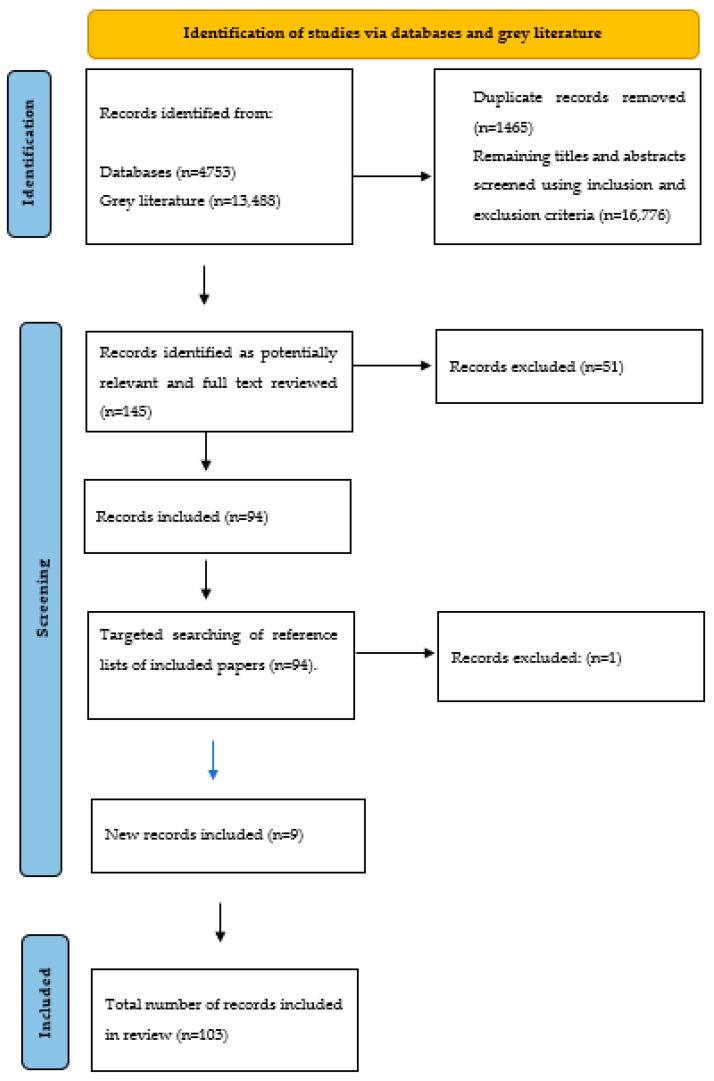
Flowchart of the review process [163].

**Table 1 ijerph-19-03450-t001:** Overview of the 103 papers included in the rapid review.

Author	Title	Year	Scope
Akkan and Canbazer [10]	The Long-Term Care response to COVID-19 in Turkey	2020	Policy paper highlighting Turkey’s response to the COVID-19 pandemic.
Anderson et al. [11]	Nursing home design and COVID-19: Balancing infection control, quality of life, and resilience	2020	Special article to discuss the need for care homes to examine architectural design models.
Arling and Arling [12]	COVID-19 and long-term care in the US State of Minnesota	2020	Policy paper highlighting US State of Minnesota’s response to the COVID-19 pandemic.
Arlotti et al. [13]	MC COVID-19 Governmental response to the COVID-19 pandemic in long-term care residences for older people: preparedness, responses and challenges for the future. Italy	2021	Policy paper highlighting Italy’s response to the COVID-19 pandemic.
Ayalon [15]	Long-term care settings in the times of COVID-19: Challenges and future directions	2020	Commentary on the challenges experienced in care homes during COVID-19.
Australian Government [14]	CASE STUDY: Dorothy Henderson Lodge	2020a	Case study example of a care home’s battle against COVID-19.
Baron-Garcia et al. [16]	Measures adopted against COVID-19 in Long-Term Care services in Catalonia	2020	Policy paper highlighting Catalonia’s response to the COVID-19 pandemic.
Belmin et al. [17]	Coronavirus Disease 2019 outcomes in French nursing homes that implemented staff confinement with residents	2020	Retrospective cohort study conducted to investigate COVID-19–related outcomes in French nursing homes that implemented voluntary staff confinement with residents.
Bergman et al. [18]	Recommendations for welcoming back nursing home visitors during the COVID-19 pandemic: Results of a Delphi panel	2020	Delphi study to generate consensus guidance statements focusing on essential family caregivers and visitors.
Blain et al. [19]	August 2020 Interim EuGMS guidance to prepare European Long-Term Care Facilities for COVID-19	2020	To guide long term care facilities in preventing the entrance and spread of SARS-CoV-2.
British Geriatrics Society [6]	Managing the COVID-19 pandemic in care homes for older people	2020	Guidance developed to help care home staff and NHS staff who work with them to support residents through the pandemic.
Browne et al. [20]	Policy response to COVID-19 in Long-Term Care Facilities in Chile	2020	Policy paper highlighting Chile’s response to the COVID-19 pandemic.
Bruquetas-Callejo and Böcker Radboud [21]	MC COVID-19 Governmental response to the COVID-19 pandemic in Long-Term Care residences for older people: preparedness, responses and challenges for the future. The Netherlands	2021	Policy paper highlighting The Netherland’s response to the COVID-19 pandemic.
Burdsall et al. [22]	A unit guide to infection prevention for long-term care staff	2017	Guidance for long-term care staff on how to prevent healthcare associated infections.
Buynder et al. [23]	Guidelines for the prevention, control and public health management of influenza outbreaks in residential care facilities in Australia	2017	Guidance for residential care facilities on the management of influenza.
Campeon et al. [24]	MC COVID-19 Governmental response to the COVID-19 pandemic in long-term care residences for older people: preparedness, responses and challenges for the future: France	2021	Policy paper highlighting France’s response to the COVID-19 pandemic.
Capucha et al. [25]	MC COVID-19 Governmental response to the COVID-19 pandemic in Long-Term Care residences for older people: preparedness, responses and challenges for the future: Portugal	2021	Policy paper highlighting Portugal’s response to the COVID-19 pandemic.
Care Provider Alliance [26]	COVID-19: Visitors’ protocol. CPA Briefing for care providers	2020a	Visitors protocol to provide practical help to care homes around visiting.
Care Provider Alliance [27]	Staff movement: a risk management framework Briefing for adult care home providers	2020b	Staff briefing to provide a risk management approach that care homes can use to manage restrictions on staff movements.
Centres for Medicare and Medicaid [29]	COVID-19 long-term care facility guidance	2020	Guidance document to provide recommendations for immediate action to reduce the spread of COVID-19.
Centres for Disease Control and Prevention [28]	Coronavirus Disease 2019 (COVID-19) Preparedness Checklist for nursing homes and other long-term care settings	2019	Guidance document to provide a checklist to be used as one tool in developing a comprehensive COVID-19 response plan.
Charlesworth and Low [30]	The Long-Term Care COVID-19 situation in Australia	2020	Policy paper highlighting Australia’s response to the COVID-19 pandemic.
Chen et al. [31]	Long-term care, residential facilities, and COVID-19: An overview of federal and state policy responses	2020	Special article to provide an overview of federal regulatory changes and state-level executive orders in relation to COVID-19.
Chuang et al. [32]	Infection control intervention methicillinlin resistant Staphylococcus aureus transmission in residential care homes for the elderly	2015	Two-arm cluster randomised controlled trial to evaluate the effectiveness of an infection control bundle in controlling methicillin-resistant Staphylococcus aureus (MRSA) transmission in residential care.
Comas-Herrera et al. [33]	International examples of measures to prevent and manage COVID-19 outbreaks in residential care and nursing home settings	2020a	Policy paper to provide examples of policy and practice measures that have been adopted internationally to prevent COVID-19 infections in care homes and to mitigate their impact.
Comas-Herrera et al. [34]	The COVID-19 long-term care situation in England	2020b	Policy paper highlighting England’s response to the COVID-19 pandemic.
Comas-Herrera et al. [35]	Rapid review of the evidence on impacts of visiting policies in care homes during the COVID-19 pandemic (Pre-print, not yet reviewed)	2020c	Rapid review on visiting policies in care homes during the COVID-19 pandemic.
Communicable Diseases Network Australia [36]	Coronavirus Disease 2019 (COVID-19) outbreaks in residential care facilities	2020	Guidelines for the control of COVID-19 outbreaks in residential care.
Da Mata and Oliveira [37]	COVID-19 and long-term care in Brazil: Impact, measures and lessons learned	2020	Policy paper highlighting Brazil’s response to the COVID-19 pandemic.
Department of Health Northern Ireland [40]	COVID-19: Guidance for nursing and residential care homes in Northern Ireland	2020	Guidance document for nursing and residential care homes.
Department of Health and Social Care [38]	Adult Social Care—our COVID-19 Winter Plan 2020/21	2020	Guidance document to provide the key elements of national support available for the social care sector for winter 2020/21.
Department of Health and Social Care [39]	Overview of adult social care guidance on coronavirus (COVID-19)	2021	Information for adult social care providers on COVID-19 guidance and support.
Directors of Adult Social Services [41]	Advice note: Cohorting, zoning and isolation practice—Commissioning for resilient care home provision	2020a	Advice note developed to support local decision making in relation to cohorting, zoning and isolation.
Directors of Adult Social Services [42]	Care homes infection control: Top tips guide	2020b	Guidance document to highlight some of the ways in which residential and nursing homes have responded to the COVID-19 pandemic.
Dumyati et al. [43]	Challenges and strategies for prevention of multidrug-resistant organism transmission in nursing homes	2017	Discussion of the challenges and potential solutions to support implementing effective infection prevention and control practices in nursing homes.
Ehrlich et al. [44]	The need for actions to protect our geriatrics and maintain proper care at U.S. long-term care facilities	2020	Discussion paper on maintaining care in US long-term care facilities.
European Centre for Disease Prevention [45]	Infection prevention and control and preparedness for COVID-19 in healthcare settings	2020	Guidance document for healthcare facilities and providers in the EU and UK on preparedness and infection prevention and control (IPC) measures for the management of COVID-19.
Fewster et al. [46]	Care homes strategy for infection prevention and control of COVID-19 based on clear delineation of risk zones	2020	Guidance document to provide a care homes strategy for infection prevention and control of COVID-19.
Forma et al. [47]	COVID-19 and clients of long-term care in Finland—impact and measures to control the virus	2020	Policy paper highlighting Finland’s response to the COVID-19 pandemic.
Glendinning [48]	MC COVID-19 Governmental response to the COVID-19 pandemic in long-term care residences for older people: preparedness, responses and challenges for the future. England	2021	Policy paper highlighting England’s response to the COVID-19 pandemic.
Gordon et al. [7]	Commentary: COVID in care homes—challenges and dilemmas in healthcare delivery	2020	To provide a commentary on challenges and dilemmas identified in the response to COVID-19 for care homes and their residents.
Gould [49]	MRSA: implications for hospitals and nursing homes	2011a	Discussion paper to update healthcare professionals’ understanding of the implications of methicillin-resistant Staphylococcus aureus(MRSA) for patients in hospital and residents in nursing homes.
Gould [50]	The challenges of caring for patients with influenza	2011b	Discussion paper to provide an overview of the nature of influenza and the challenges that it poses to older people and those who care for them.
Government of Canada [51]	Infection prevention and control for COVID-19: Interim guidance for long-term care homes	2021	Guidance on infection prevention and control for COVID-19.
Graham and Wong [52]	Responding to COVID-19 in residential care: The Singapore experience	2020	Policy paper highlighting Singapore’s response to the COVID-19 pandemic.
Griffin [53]	COVID-19: Experts urge strategies to prevent further outbreaks in care homes	2020	News article (BMJ).
Haenen et al. [54]	Surveillance of infections in long-term care facilities (LTCFs): The impact of participation during multiple years on health care-associated infection incidence	2019	Analysis of trends in data gathered by a national sentinel surveillance network.
Harold Van Houtven et al. [55]	Impact of the COVID-19 outbreak on long-term care in the United States	2020	Policy paper highlighting the US’s response to the COVID-19 pandemic.
Hasmuk et al. [56]	The long-term care COVID-19 situation in Malaysia	2020	Policy paper highlighting Malaysia’s response to the COVID-19 pandemic.
Health Protection Scotland [57]	COVID-19: Information and guidance for care home settings (adults and older people)	2020	Guidance for all services registered with the Care Inspectorate as care homes, except those for children and young people.
Health Information and Quality Authority [110]	Rapid review of public health guidance for residential care facilities in the context of COVID-19	2021	A rapid review of public health guidance for residential care facilities in the context of COVID-19.
Heudorf et al. [58]	COVID-19 in long-term care facilities in Frankfurt am Main, Germany: Incidence, case reports, and lessons learned	2020	To discuss the experiences with COVID-19 in nursing homes in Frankfurt.
Higginson [59]	Minimising infection spread of influenza in the care setting	2018	Discussion paper to outline transmission risks and infection prevention of influenza.
Hsu et al. [61]	Understanding the impact of COVID-19 on residents of Canada’s long-term care homes—ongoing challenges and policy responses	2020	Policy paper highlighting Canada’s response to the COVID-19 pandemic.
Hurley and Burke [60]	MC COVID-19 Governmental response to the COVID-19 pandemic in long-term care residences for older people: Preparedness, responses, and challenges for the future. Ireland	2021	Policy paper highlighting Ireland’s response to the COVID-19 pandemic.
Jacobs et al. [62]	The Impact of COVID-19 on long-term care facilities in South Africa with a specific focus on dementia care	2020	Policy paper highlighting South Africa’s response to the COVID-19 pandemic, with a specific focus on dementia care.
Johansson and Schön [63]	MC COVID-19 Governmental response to the COVID-19 pandemic in long-term care residences for older people: preparedness, responses and challenges for the future. Sweden	2021	Policy paper highlighting Sweden’s response to the COVID-19 pandemic.
Jump and Donskey [64]	Clostridium difficile in the long-term care facility: Prevention and management	2015	To discuss epidemiology and management of C. difficile infection among residents of long-term care facilities.
Kim [65]	The impact of COVID-19 on long-term care in South Korea and measures to address it	2020	Policy paper highlighting South Korea’s response to the COVID-19 pandemic.
Koshkouei et al. [66]	How can pandemic spreads be contained in care homes?	2020	Rapid review evaluating available measures to minimise the risk of infection spread within care home settings.
Kruse et al. [67]	The impact of COVID-19 on long-term care in the Netherlands: the second wave.	2020	Policy paper highlighting the Netherland’s response to the COVID-19 pandemic.
Lorenz-Dant [68]	Germany and the COVID-19 long-term care situation	2020	Policy paper highlighting Germany’s response to the COVID-19 pandemic.
Low et al. [69]	Safe visiting at care homes during COVID-19: A review of international guidelines and emerging practices during the COVID-19 pandemic	2021	Policy paper exploring guidelines for safe visiting at care homes during the COVID-19 pandemic.
Lückenbach et al. [70]	MC COVID-19 Governmental response to the COVID-19 pandemic in long-term care residences for older people: preparedness, responses and challenges for the future. Germany	2021	Policy paper highlighting Germany’s response to the COVID-19 pandemic.
Manitoba Health [71]	Routine practices and additional precautions: Preventing the transmission of infection in health care	2019	Guidance document on preventing the transmission of infection in health care
Ma’u et al. [72]	COVID-19 and long-term care in Aotearoa New Zealand	2020	Policy paper highlighting New Zealand’s response to the COVID-19 pandemic.
McGilton et al. [73]	Uncovering the devaluation of nursing home staff during COVID-19: Are we fuelling the next health care crisis?	2020	Editorial.
Ministry of Health and Long Term Care [74]	Control of respiratory infection outbreaks in long-term care homes, 2018	2018	Guidance document on controlling respiratory infection outbreaks in care homes.
Minnesota Dept of Health [75]	COVID-19 Toolkit: Information for long-term care facilities	2020	Toolkit for long-term care facilities.
Montoya et al. [76]	Infections in nursing homes: Epidemiology and prevention programs	2016	Discussion paper.
National Collaborating Centre for Methods and Tools [77]	Rapid Review: What risk factors are associated with COVID-19 outbreaks and mortality in long-term care facilities and what strategies mitigate risk?	2020	Rapid review on risk factors for COVID-19 outbreaks.
Nazarko [78]	How to control the risk of MRSA infection	2006	To discuss the effects of MRSA and the ways in which care home workers can prevent its spread.
NHS Shetland [79]	Procedure for the prevention control and management of Clostridium Difficile infection in care settings in Shetland	2017	Guidance document for healthcare settings in Shetland.
Public Health Agency Canada [80]	Guidance: Infection prevention and control measures for healthcare workers in acute care and long-term care settings seasonal influenza	2010	Guidance document on infection prevention and control measures for seasonal influenza.
Rajan and McKee [81]	Learning from the impacts of COVID-19 on care homes: A pilot survey	2020	Pilot study to establish the impacts of COVID-19 on care homes.
Rios et al. [82]	Guidelines for preventing respiratory illness in older adults aged 60 years and above living in long-term care	2020a	To identify infection protection and control recommendations from published clinical practice guidelines (CPGs) for adults aged 60 years and older in long-term care settings.
Rios et al. [83]	Preventing the transmission of COVID-19 and other coronaviruses in older adults aged 60 years and above living in long-term care: a rapid review	2020b	To examine the current guidelines for infection prevention and control of coronavirus disease-19 (COVID-19) or other coronaviruses in adults 60 years or older living in long-term care facilities (LTCF).
Schmidt et al. [84]	The impact of COVID-19 on users and providers of long-term care services in Austria	2020	Policy paper highlighting Austria’s response to the COVID-19 pandemic.
Scopetti et al. [85]	Expanding frontiers of risk management: care safety in nursing home during COVID-19 pandemic	2021	Discussion paper on care safety in nursing homes during the COVID-19 pandemic.
Shallcross et al. [86]	Risk factors associated with SARS-CoV-2 infection and outbreaks in Long Term Care Facilities in England: a national survey	2020	Cross-sectional survey to identify risk factors for SARS-CoV-2 infection and outbreaks in long-term care facilities (pre peer-review manuscript).
Shi et al. [87]	Report from mainland China: Policies to support long term care during the COVID-19 outbreak	2020	Policy paper highlighting policies to support long term care during the COVID-19 pandemic.
Shrader et al. [88]	Responding to a COVID-19 outbreak at a long-term care facility	2021	Describes an outbreak of COVID-19 in a long-term care facility (LTCF) in West Virginia that was the epicentre of the state’s pandemic.
Smith et al. [89]	SHEA/APIC guideline: Infection prevention and control in the long-term care facility	2008	Discussion of infection prevention and control guidelines in long-term care facilities.
Spilsbury et al. [90]	Less COVID-19: Learning by experience and supporting the care home sector during the COVID-19 pandemic: Key lessons learnt, so far, by frontline care home and NHS staff	2020	Interview study capturing the experiences of frontline care home and NHS staff caring for older people with COVID-19.
Stall et al. [91]	Finding the right balance: An evidence-informed guidance document to support the re-opening of Canadian nursing homes to family caregivers and visitors during the Coronavirus disease 2019 pandemic	2020	Guidance document on re-opening of Canadian nursing homes to family caregivers and visitors during the COVID-19 pandemic.
Stanwell-Smith [92]	Advice for the influenza season: Infection control in the care home	2008	Discussion paper on the methods of infection control for influenza and other acute upper respiratory infections.
Suarez-Gonzalez et al. [93]	The impact of COVID-19 in the Long-term care system in Asturias (Spain)	2020a	Policy paper highlighting Austria’s response to the COVID-19 pandemic.
Suarez-Gonzalez [94]	Detrimental effects of confinement and isolation on the cognitive and psychological health of people living with dementia during COVID-19: emerging evidence	2020b	To describe the effects of lockdown on people with dementia.
Szebehely [95]	The impact of COVID-19 on long-term care in Sweden	2020	Policy paper highlighting Sweden’s response to the COVID-19 pandemic.
Urbé [96]	MC COVID-19 Governmental response to the COVID-19 pandemic in long-term care residences for older people: preparedness, responses, and challenges for the future	2021	Policy paper highlighting Luxembourg’s response to the COVID-19 pandemic.
Victoria State Government [98]	Respiratory illness in residential and aged care facilities: Guidelines and information	2018	Guidance document on respiratory illness in residential and aged care facilities.
Verbeek et al. [97]	Allowing visitors back in the nursing home during the COVID-19 crisis: A Dutch national study into first experiences and impact on well-being	2020	Mixed-methods cross-sectional study on visiting care homes during the COVID-19 pandemic.
Wang [99]	Use the environment to prevent and control COVID-19 in senior-living facilities: An analysis of the guidelines used in China	2020a	Content analysis of the guidelines on COVID-19 control issued by the State Council of China in February 2020 for senior-living facilities.
Wang et al. [100]	Prevention and control of COVID-19 in nursing homes, orphanages, and prisons.	2020b	Discussion paper on prevention and control strategies for COVID-19.
Winfield and Wiley [101]	Tackling infection in care homes	2012	Discussion paper describing a three-dimensional strategy that reduced MRSA colonisation.
Wong et al. [102]	The COVID-19 long-term care situation in Hong Kong: Impact and measures	2020	Policy paper highlighting Hong Kong’s response to the COVID-19 pandemic.
World Health Organisation [103]	Preventing and managing COVID-19 across long-term care services	2020a	Policy paper providing objectives and key action points to prevent and manage COVID-19 across long-term care.
World Health Organisation [104]	COVID-19 Infection prevention and control. Communication toolkit for long-term care facilities	2020b	Communication toolkit to protect residents and staff from infection and prevent potential spread of COVID-19 and other respiratory pathogens within long-term care facilities.
World Health Organisation [105]	COVID-19 Infection prevention and control. Preparedness checklist for long-term care facilities	2020c	Checklist to be used by facility administrators, IPC focal points or staff, and internal or external professionals.
World Health Organisation [106]	Guidance on COVID-19 for the care of older people and people living in long-term care facilities, other non-acute care facilities and home care	2020d	Guidance on COVID-19 for the care of older people and people living in long-term care facilities.
World Health Organisation [107]	Infection prevention and control guidance for long-term care facilities in the context of COVID-19	2020e	Interim guidance document on infection prevention and control of COVID-19.
Ylinen et al. [108]	MC COVID-19 Governmental response to the COVID-19 pandemic in long-term care residences for older people: Preparedness, responses and challenges for the future	2021	Policy paper highlighting Finland’s response to the COVID-19 pandemic.
Zalakaín and Davey [109]	The COVID-19 on users of long-term care services in Spain	2020	Policy paper highlighting Spain’s response to the COVID-19 pandemic.

**Table 2 ijerph-19-03450-t002:** Challenges and facilitators of implementing COVID-19 interventions in care homes.

Challenges	Facilitators
**Staffing and workload** The COVID-19 pandemic exacerbated pre-existing problems in care homes, such as staff shortages/sickness; lack of resources, training and equipment; and excessive workloads, and this made the implementation of COVID-19 interventions more difficult [7,24,44,47,55,60,61,62,67,68,70,72,73,90,95,103,109].	**Sufficient staff** support Having sufficiently supported staff helped facilitate the successful implementation of COVID-19 interventions. This included ensuring staff were paid for any time spent in isolation [34,48,81,107]; rewarding staff with annual pay increases or bonuses, gifts, care packages or additional leave days [34,52,61,81]; providing food and water stations to ensure staff were adequately fed and hydrated or providing access to wellbeing initiatives, counselling, and emotional support [42,52,81].
**Lack of guidance/clarity from governments** Implementing COVID-19 interventions was made more difficult by the lack of guidance and clarity from governments around when and how interventions should be applied, with policy measures often scarce, flawed or adopted late [21,24,30,34,55,60,62,81,95,108].	**Good communication** Having good communication and the availability of informational materials, such as brochures, posters and signage on COVID-19 and the associated policies helped explain the reasons behind COVID-19 interventions to residents and their families and friends [14,28,36,40,75,104].
**Physical space and layout of care homes** The physical space and layout of some care homes made implementing COVID-19 interventions more difficult. For example, not all care homes had the space to provide single rooms, to create separate ‘zones’ or to ensure sufficient walking space around the home in line with social distancing measures [11,38,42,56,60,62,81,90,103,109].	**Use of innovative technology** Innovative technology and software such as Zoom, Facetime or Teams helped remotely support residents and their families during periods of restriction and reduce the impact of social isolation [33,42,44,60,90,99]. However, some problems were highlighted around having insufficient equipment, broadband or Wi-Fi within care homes [30]; as well as the requirement for staff, families and residents to have training on how to use the technology [107].

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
