# Peer review of "Social Distancing and Isolation Strategies to Prevent and Control the Transmission of COVID-19 and Other Infectious Diseases in Care Homes for Older People: An International Review"

_ijerph, 2022, doi:10.3390/ijerph19063450_

Round 1

Reviewer 1 Report

Thank you very much for this opportunity to review this paper. This article is especially interesting in that the topic is relevant to this era of COVID-pandemic. My comments regarding the paper is as below.

  1. Overall, the authors summarized the prevention measure at CHs. Although summarizing the preventive measures are important, the information is already widely available and the importance of conducting this review at this stage is not convincing. What the reader what to know may not be the summary of the prevention measure, which are already be accessible in report or document format. A new layer of summary or point of view should be added. 
  2. The title is the 'effectiveness' of the prevention measure however, the authors state that there are no empirical evidence. The wording should be revised. 
  3. The sub-titles in the Findings sections are confusing. For example, Social distancing at LINE 206 should be in larger font or be numbered at the top to prevent confusion (if I am reading the paper correctly.

Reviewer 2 Report

1.The authors need to state briefly about the process for study selection, data extraction, and data synthesis followed Cochrane evidence-informed guidance for conducting rapid reviews. (lines64-66)

2.Please describe the meaning of the works of the four reviewers. (lines104-107)

3.The criteria for the reviewers to appraise the papers need to be described.

4.Figure 1:Records identified as potentially relevant and full text reviewed. This column needs to add (n=145).

Round 2

Reviewer 1 Report

Thank you very much for this opportunity to revise the paper. 

  1. The authors have clearly emphasized the importance of the work in the paper.
  2. They have also edited the title, and the misleading subtitles. 

After some final grammatical revisions, I believe this paper is ready for publication.